# Characteristics of the First Protein Tyrosine Phosphatase with Phytase Activity from a Soil Metagenome

**DOI:** 10.3390/genes10020101

**Published:** 2019-01-29

**Authors:** Genis Andrés Castillo Villamizar, Heiko Nacke, Laura Griese, Lydia Tabernero, Katrina Funkner, Rolf Daniel

**Affiliations:** 1Department of Genomic and Applied Microbiology and Göttingen Genomics Laboratory, Institute of Microbiology and Genetics, Georg-August University of Göttingen, Grisebachstr. 8, 37077 Göttingen, Germany; gcastil@gwdg.de (G.A.C.V.); hnacke@gwdg.de (H.N.); lauragriese@live.de (L.G.); katrina.funkner@stud.uni-goettingen.de (K.F.); 2Línea Tecnológica Biocorrosión, Corporación para la investigación de la corrosión C.I.C., Piedecuesta 681011, Santander, Colombia; 3School of Biological Sciences, Faculty of Biology, Medicine and Health, University of Manchester, Manchester M13 9PL, UK; lydia.tabernero@manchester.ac.uk

**Keywords:** metagenomics, phosphatases, phytases, promiscuous enzymes, metagenomic library

## Abstract

Protein tyrosine phosphatases (PTPs) fulfil multiple key regulatory functions. Within the group of PTPs, the atypical lipid phosphatases (ALPs) are known for their role as virulence factors associated with human pathogens. Another group of PTPs, which is capable of using inositol-hexakisphosphate (InsP_6_) as substrate, are known as phytases. Phytases play major roles in the environmental phosphorus cycle, biotechnology, and pathogenesis. So far, all functionally characterized PTPs, including ALPs and PTP-phytases, have been derived exclusively from isolated microorganisms. In this study, screening of a soil-derived metagenomic library resulted in identification of a gene (*pho16B*), encoding a PTP, which shares structural characteristics with the ALPs. In addition, the characterization of the gene product (Pho16B) revealed the capability of the protein to use InsP_6_ as substrate, and the potential of soil as a source of phytases with so far unknown characteristics. Thus, Pho16B represents the first functional environmentally derived PTP-phytase. The enzyme has a molecular mass of 38 kDa. The enzyme is promiscuous, showing highest activity and affinity toward naphthyl phosphate (K_m_ 0.966 mM). Pho16B contains the HCXXGKDR[TA]G submotif of PTP-ALPs, and it is structurally related to PtpB of *Mycobacterium tuberculosis*. This study demonstrates the presence and functionality of an environmental gene codifying a PTP-phytase homologous to enzymes closely associated to bacterial pathogenicity.

## 1. Introduction

Since the emergence of next generation sequencing and omics approaches, the genetic material of numerous organisms, including bacterial, plant, and animal pathogens, has been analyzed. Genome analysis of single organisms, together with metagenomic and metaproteomic surveys comprising diverse environmental samples, provided an improved understanding of microbial biodiversity and the relationship of diversity with ecological, biotechnological, evolutionary or pathogenic processes [1,2,3].

Protein tyrosine phosphatases (PTPs) are an example of enzymes associated with virulence whose environmental homologues have not fully been studied. PTPs have multiple roles in cell metabolism. For many PTPs, their physiological substrate has not been identified [4]. A major role of the PTPs includes regulating, together with the protein kinases, the cellular equilibrium of protein tyrosine phosphorylation by dephosphorylating tyrosine residues of protein substrates. PTPs also participate in cell signaling by dephosphorylating proteins on other amino acidic residues (serine and threonine), as well as lipid substrates [4,5]. Another group of PTP proteins, known as atypical lipid phosphatases (ALPs), are associated with different levels of metabolic control of phosphoinositides (PIs) [6]. ALPs possess a characteristic catalytic profile, sequence, and domain organization. They harbor a distinct active site P-loop signature (HCXXGKDR[TA]G), containing the acid/base catalyst (D) and an extra basic residue (K) important in substrate binding [6,7]. PTP-ALPs with this P-loop are found in bacteria, including human pathogens [6], but only very few have been characterized.

Other members of the PTPs can hydrolyze *myo*-inositol phosphates (InsPs), which are ubiquitous products of inositol metabolism and bear a high level of structural resemblance to PIs [8]. The phytic acids (*myo*-inositol hexakisphosphate, InsP_6_) have several roles within eukaryotic cells, including second messengers and cofactors that facilitate the regulation of diverse biochemical processes, such as transcription and hormone receptor activity [9,10]. A variety of other important biological functions have been directly or indirectly related to the presence of InsP_6_. It has been reported that InsP_6_ acts as a signal in the maintenance of basal resistance to viruses and phytopathogens [11]. Other reported functions of InsP_6_ include antioxidative functions or involvement in DNA repair in prostate cancer prevention [12,13]. InsPs are typically absent in prokaryotes, but several genes encoding phosphatases (phytases) capable of using InsP_6_ as substrate have been detected in bacteria [14,15,16].

InsP_6_ also represents the main phosphorus (P) storage in many types of plants. It is considered a significant part of organic soil phosphate (Po) and relevant for the P cycle in soils [17]. Phytases decompose InsP_6_ to less phosphorylated *myo*-inositol derivatives and inorganic phosphate [18]. Phytases are used as an effective feed additive that increases the digestion/absorption rates of phosphorus in cereal-based feed. In this way, livestock growth increases and phosphorus pollution caused by the non-assimilated InsP_6_ are reduced [19]. The annual phytase market value is roughly $700 million. Therefore, the search for new phytases has become a major research challenge [20]. However, few of the phytases—all derived from cultured microorganisms—have been commercially exploited.

Currently, four classes of phytases have been described: histidine acid phytase (HAPs-phy), β-propeller phytase (BPPs-phy), purple acid phytase (PAPs-phy), and protein tyrosine phytase (PTPs-phy). These enzymes are not structurally similar and use different mechanisms to cleave phosphate groups from InsP_6_ [18,21,22]. PTPs have been relatively well studied, but not many enzymes of this type with phytase activity have been reported. Moreover, most of these phytases are derived from culturable anaerobic bacteria [21,23].

Several functional gene homologs of physiologically relevant kinases/phosphatases have been found as ubiquitous in non-human environments, demonstrating the relevant ecological roles of these genes for the activity and survival of environmental bacteria [3]. Although homologous sequences of PTPs, including ALPs, are present in the genome sequences of numerous microorganisms (cultured and uncultured), to our knowledge, characterized PTPs from environmental samples have not been reported. In addition, PTPs carrying the specific P-loop of the ALPs with phytase activity are also unknown. Here, we report, for the first time, a functional metagenome-derived PTP protein carrying the specific P-loop of ALPs and exhibiting phytase activity.

## 2. Results and Discussion

### 2.1. Identification and In Silico Characterization of the Novel ALP Member Pho16B

Most of the research on environmental phytases is represented by a few studies performing sequence-based identification of putative phytase genes in metagenomes. A limited number of the corresponding phytase proteins have been characterized. In addition, only three phytase-encoding genes derived from functional metagenomic approaches have been reported [24,25,26,27]. We have previously reported, to date, the highest number of environmentally derived phosphatases [28]. Function-based screening of a forest soil metagenomic library for genes coding for phosphatases and phytases yielded a positive *Escherichia coli* clone carrying the recombinant plasmid pLP16. The insert of this plasmid (7806 bp) harbors 40 open reading frames (ORFs). For 20 of the detected ORFs, it was possible to assign a putative function by similarity searches (Figure 1, Appendix A).

The taxonomic classification of the complete pLP16 insert sequence is affiliated to the Gram-positive soil bacterium *Streptacidiphilus melanogenes*, originally isolated from *Pinus*-associated soils [29], and other Actinobacteria. Therefore, an actinobacterial relative of *S. melanogenes* might represent the original source of the cloned fragment. Consistently, the deduced protein sequence-based analysis showed that the gene product of *pho16B*, encoding a putative phytase, is closely related to the PTP WP_042381880 from *S. melanogenes* (79% of sequence identity). In addition to protein sequences stored in the National Center for Biotechnology Information (NCBI) nr database, the Pho16B sequence was compared with protein sequences of specialized metagenome databases. The closest related match to Pho16B in the latter databases was entry MGYP000356208135 of the European Bioinformatics Institute (EMBL-EBI) metagenomics platform. MGYP000356208135 shows 42% sequence identity to Pho16B. The corresponding gene sequence was derived from a sludge metagenome associated to an oil refinery and is similar to a gene encoding a PTP from *Sphingomonas* sp. (UniProt entry A0A1M2ZKN6).

The sequence analysis of *pho16B* revealed that it encodes a PTP, which contains the characteristic ALP signature (HCXXGKDR[TA]G). The presence of the conserved Lys and Asp residues indicates similarity of the ALPs and Pho16B with respect to the predicted catalytic mechanism [6,7] (Figure 2).

Pho16B contains a signal peptide with a predicted cleavage site between amino acid positions 36 and 37, indicating secretion of the protein and extracellular localization in the original host. Pho16B carries the PTPc domain and the specific Y_phosphatase3 domain (Figure 2A). The *pho16B* gene was subcloned in the expression vector pBAD202/D-TOPO and expressed in the *E. coli* host strain LMG194. Subsequently, the *pho16B* gene product was purified by a combination of affinity and hydrophobic interaction chromatography, yielding 155 µg pure enzyme from 500 mL culture, with a specific activity of 8.09 U/mg. The protein has a calculated molecular mass of approx. 38 kDa, which is similar to the molecular masses of other PTP-ALPs, such as lmo1800 and lmo1935 from *Listeria monocytogenes*, and phytases like Bd1204 of *Bdellovibrio bacteriovorus* [30].

We analyzed the phylogenetic position of Pho16B (Figure 3) by considering the phosphatase groups (G1–G9) established by Beresford et al. 2010 [6]. Bacterial sequences are clustered in the groups G1 to G6, while the eukaryotic sequences are clustered in the groups G7 to G9. The P96830 (MptpB protein of *Mycobacterium tuberculosis*) belongs to G1 together with other mycobacterial ALPs, whereas Pho16B belongs to G4, which comprises phosphatases of Gram-positive and Gram-negative bacteria. The closest related protein of Pho16B was included in our phylogenetic analysis. This protein, belonging also to G4, was annotated during genome analysis of *S. melanogenes* (accession WP_042381880). To our knowledge, the expression or characterization of the gene product have not been reported.

### 2.2. Tertiary Structure Prediction of Pho16B

In addition to the sequence comparisons, we further predicted the theoretical 3D model of Pho16B, to determine its closest structural relative. This analysis was performed by using the I-TASSER tools [31]. For the modeling, I-TASSER initially uses different threading programs generating tens of thousands of template alignments. Only the templates of the highest significance in the threading alignments were used. The significance of the alignments was measured by a normalized Z-score. Z-scores > 1 mean a good alignment and vice versa (Appendix A).

Employment of I -TASSER suite resulted in the prediction of five models for Pho16B. Two of the predicted models (M1 and M2) showed C-score values higher than −1.5 (−1.22 and −1.36, respectively). Therefore, the model M1 was selected as the most plausible model for Pho16B (Appendix A). The model M1 was related to the Protein Data Bank (PDB) entry 1YWF, that corresponds to the crystal structure of *M. tuberculosis* PTP (PtpB). It is assumed that a C-score higher than −1.5 confirms a correct global topology [31]. As the native structure of Pho16B is not known, the quality of the modeling prediction is determined by calculating the distance between the predicted model and published native structures. In our case, I-TASSER predicted the quality of the model by calculating the TM-score. A TM-score with a value of 0.56 was calculated, indicating a very similar fold of the Pho16B and the reference protein PtpB [32]. The analysis of the occurrence and distribution of clinically relevant bacterial virulence genes across natural (non-human) environments has demonstrated the presence of transcribing homologs of several virulence genes in those habitats. Consequently, a common ancestral origin between the environmental genes and their virulence-associated counterparts has been established [3]. Pho16B is the first reported environmental PTP with phytase activity, carrying a homologous molecular signature of virulence gene-related products (ALPs). Therefore, this study provides insights into the structural similarities of environmental PTPs to other known phosphatases related to virulence factors of human pathogens.

### 2.3. Optimum Temperature and pH of Pho16B

The characterization of the biochemical properties of Pho16B revealed its optimal activity to occur at a temperature of 45 °C. More than 50% of its activity was lost at temperatures ≥ 50 °C (Figure 4A). In general, phytases show activity within a wide range of temperatures, and some previously reported bacterial representatives are affected by temperature in a similar way as Pho16B. The phytase of *B. bacteriovorus* Bd1204 exhibits optimal activity at a slightly higher temperature than Pho16B, but also loses most of its activity (40%) at temperatures higher than 50 °C [30].

The optimal pH of Pho16B enzymatic activity (5.6) was detected by measurements across a set of overlapping buffer systems ranging from pH 2 to 9. At pH 4.0 and neutral pH, Pho16B retained 80% and 70% of the activity, respectively. At slightly alkaline pH value of 7.5, activity was reduced to 50% (Figure 4B). The activities of other reported PTP phytases show a strong reduction at this pH level, i.e., the enzyme phyAme from *Megasphaera elsdenii* loses its activity almost completely at pH 7.0 [33]. The optimal activity and higher capacity of enzymatic activity retention of Pho16B at low pH match with the predicted extracellular localization of the enzyme at its natural acidic forest soil habitat, which has a soil pH of 3.5 [34,35]. This is also consistent with the cysteine-based catalytic mechanism of MptpB and other PTPs, whose optimal catalytic activities are at acidic pH values [36].

### 2.4. Pho16B Converts a Broad Range of Substrates

The substrate specificity of Pho16B was tested using ten different substrates (Figure 5). Pho16B showed activity in the presence of all tested compounds. The highest relative activity was detected with naphthyl phosphate, and the lowest with NADP as substrate. The relative activity with InsP_6_ was approximately 30% of that with naphthyl phosphate (Figure 5). This indicated that InsP_6_ is not the primary substrate of Pho16B under the tested conditions. Nevertheless, to our knowledge, this is the first reported example of an environmental PTP with activity towards InsP_6_. We previously demonstrated the phytase activity of environmental phosphatases such as alkaline phosphatases and acid phosphatases [28]. Many phosphatases are recognized as promiscuous types of enzymes [37]. The broad substrate spectrum indicated that Pho16B belongs also to this type.

The kinetic parameters of Pho16B were determined using the purified protein and InsP_6_, naphthyl phosphate, and *p*-nitrophenylphosphate as substrates under optimal pH and temperature conditions (pH 5.6 and 45 °C; Table 1). Pho16B shows a higher affinity and catalytic efficiency for naphthyl phosphate than for InsP_6_. The protein tyrosine phosphatase (PhyAsr) from *Selenomonas ruminantium*, which is one of the few characterized and reported (PTP)-like phytases, possesses similar affinity (*K_m_*) and catalytic efficiency (*K_cat_*/*K_m_*) towards InsP_6_ under high ionic strength conditions (1 mM and 163 mM^−1^·s^−1^, respectively) [38]. The protein tyrosine phosphatase of *Yersinia* Yop51, which is involved in pathogenesis, displays affinity values almost three times lower than Pho16B with *p*-nitrophenylphosphate as substrate (*K_m_* 2.90 mM). However, the turnover number of Yop51 (1235 ± 36 s^−1^) is significantly higher than the one of Pho16B [39]. Protein phosphatases, including phytases, possess a wide range of catalytic traits. In this sense, the analyzed kinetic characteristics of Pho16B are in the normal range compared with the currently reported phytases [14,18,39]. However, since Pho16B is the first reported environmental PTP, it is not possible to compare its kinetic parameters with those from phytases of the same type.

### 2.5. Effect of Additives on Pho16B Activity

The effects of various additives on Pho16B enzyme activity are summarized in Figure 6. Co^2+^, Cu^2+^, and Fe^2+^ inhibit enzyme activity by more than 50%. In the presence of Al^3+^, no activity was detected. It has been previously demonstrated that the phosphatase activity of soil-derived enzymes might be inhibited in the presence of aluminum hydroxides [40]. Moreover, Cu^2+^ and Fe^2+^ have been reported as strong inhibitors of several phosphatases and phytases from different organisms, such as *S. ruminantium* or *Klebsiella terrigena* [41,42]. These proteins were also strongly inhibited in the presence of Zn^2+^, which only has a minor effect on the activity of Pho16B. The presence of Co^2+^ reduced the relative activity of Pho16B by 80%. By contrast, other phosphatases, i.e., the enzymes derived from bovine rumen bacterium *Mitsuokella jalaludinii*, are not affected by the presence of Co^2+^ [42]. The glucose-1-phosphatase with phytase activity (AgpP) from *Pantoea. agglomerans* possesses an unusual metal ion activation. Ca^2+^, Mg^2+^, and Mn^2+^ enhance the activity of this enzyme up to 200% [43]. The activity of Pho16B was not affected by the addition of Ca^2+^ or Mg^2+^, but the presence of Mn^2+^ slightly increased the enzymatic activity of Pho16B. 

The addition of dithiothreitol (DTT) had a very strong inhibitory effect, depleting 80% of Pho16B activity. Other previously reported phosphatases did not show significant losses of activity in the presence of this reagent [44,45]. It has been shown that DTT can reduce protein function by disrupting disulfide bonds of the proteins or acting as a metal ion chelator [46]. Another possibility is that DTT could inhibit Pho16B activity by competing with its substrates. It has been proposed that DTT can interact with the catalytic domain of the enzymes by steric hindrance via hydrogen interactions with amino acid residues [47]. Wolframate and oxalate also reduce the activity of Pho16B to less than 60%. Oxalate is known as an inhibitor of acid phosphatase activity [48], and wolframate as inhibitor of phosphatase activity of PTPs from plants [49].

## 3. Materials and Methods

### 3.1. Sampling, Metagenomic Library Construction, and Function-Based Screening

The enzyme Pho16B originated from an Arenosol soil A horizon sample (SEW2) with a pH of 3.46, which was collected from a beech forest site within the Schorfheide-Chorin biosphere reserve in Germany. Collection of the sample was performed as previously described by Kaiser et al. (2016) [50]. The metagenomic library (SEW2) comprised 135,240 *E. coli* clones and was constructed using the plasmid pCR-XL-TOPO as vector (Invitrogen GmbH, Karlsruhe, Germany) and the method described by Nacke et al. [35]. The previous screening of the metagenomic library was performed by following the method of Villamizar et al.. Phosphatase/phytase positive library-bearing *E. coli* clones growing on modified Sperber minimal medium, using InsP_6_ as phosphorous source and 5-bromo-4-chloro-3-indolyl phosphate (BCIP) as indicator, turn from white to dark blue within 48 h [51].

### 3.2. Molecular Analysis

The insert sequence of plasmid pLP16 (7972 bp), which was isolated from a metagenomic library clone showing phosphatase activity [28], was subjected to sequence analysis. Open reading frame (ORF) prediction was performed using the ORF finder tool provided by the NCBI and the ARTEMIS program [52,53]. Since the metagenomic library was constructed from metagenomic DNA fragments and cloned into the pCR-XL-TOPO vector, all genes predicted for both strands were analyzed, in order to identify the putative gene or genes responsible for the activity on the indicator plate. The results were verified and improved manually by using criteria such as the presence of a ribosome-binding site, GC frame plot analysis, and similarity to known genes. Amino acid sequences deduced from the ORFs were examined for similarities with known protein families and domains by performing searches against the InterPro collection of protein signature database and the Conserved Domain Database (CDD). Next, a search of phosphatase signatures was performed by using a local version of PhosphaBase [54,55,56]. Signal peptide prediction was performed using SIGNALP 4.0 [57]. Additionally, the taxonomic classification of the complete DNA insert was performed by using the software KAIJU [58]. Similarity searches of candidate gene product of *pho16B*, were performed by using the NCBI databases non-redundant sequences (nr) and metagenomic proteins (env_nr), and a Basic Local Alignment Search for proteins (BLASTP). A second search against metagenomic data was performed by using the metagenomics platform of the European Bioinformatics Institute [59,60].

Multiple sequence alignments of Pho16B encoded by gene *pho16B* and related phosphatases, representing the phosphatase groups previously defined by Beresford et al. [6], were performed using MUSCLE [61]. A phylogenetic consensus tree was calculated using the neighbor-joining method with MEGA X and 500 bootstrap replicates [62]. The tree was visualized using iTOL v3 [63]. The evolutionary distances were calculated using the number of differences method [3].

A prediction of the tertiary structure of protein Pho16B was performed by employing the I-TASSER software suite [64,65]. The quality of models generated using I-TASSER is based on two major criteria: the confidence score (C-score) and the template modeling score (TM-score). I-TASSER generated five models. The models were ranked based on the C-score (confidence score). The C-scores are calculated on the basis of the statistical significance of the threading profile–profile alignment, as well as structure convergence of the assembly simulations. The C-score ranged from −5 to 2. A high C-score value indicates a model with greater confidence [32].

The TM-score (the template modeling score) addresses the structural similarity of two protein models. This score can solve some difficulties of the commonly used metrics, such as the root-mean-square deviation (RMSD). The TM-score measures the global fold similarity. Moreover, TM-score is less sensitive to local structural variations. Another advantage of this measurement is that the magnitude of the TM-score for random structure pairs is length-independent. The TM-score has a value range of 0 to 1, whereby the value 1 indicates a perfect match between two structures [32]. By calculating the TM-score, we obtained an estimation about the structural similarity between the predicted model of Pho16B and published native or experimentally determined structures. Values close to 0.5 indicate a model of correct topology. In this study, the model with the highest confidence score (C-score) was selected as the best predicted optimized 3D modeling structure.

### 3.3. Biochemical Characterization of Pho16B

The ORF *pho16B* was cloned into plasmid pBAD202/D-TOPO (Thermo Fisher Scientific GmbH, Schwerte, Germany). In this way, sequences encoding the His_6_ and thioredoxin tags were added to the N terminus of Pho16B. Plasmid DNA containing the insert cloned in the correct orientation was used for transforming *E. coli* LMG194 cells. Transformants were grown on Luria-Bertani (LB) agar plates [66] supplemented with kanamycin (50 µg/µL) and incubated at 37 °C. A colony of *E. coli* LMG194 harboring the *pho16B*-pBAD202 construct was used to inoculate 500 mL M9 minimal salts medium [67] supplemented with 50 µg/µL kanamycin and 2% glycerol. The culture was incubated under shaking at 90 rpm by using a New Brunswick Innova 44 incubator-shaker (Eppendorf AG, Hamburg, Germany) at 30 °C. Protein expression was induced at an OD_600_ of 0.8 using L-arabinose (end concentration 0.02%) and incubated for 16 h.

The cells were harvested by centrifugation for 30 min at 4 °C and 8000 rpm (Sorvall^®^ RC6 centrifuge, rotor SLA 3000, Thermo Fisher Scientific). The resulting cell pellets were suspended in 10 mL 50 mM Tris, 250 mM NaCl buffer. Mechanical cell disruption was performed using a French press (1.38 × 10^8^ Pa; Thermo Fisher Scientific). Subsequently, the extract was cleared by centrifugation for 0.5 h at 4 °C and 15,000 rpm (Sorvall^®^ RC6 centrifuge with rotor SS 35, Thermo Fisher Scientific). The crude extract was filtered twice using 0.45 µm and 0.2 µm filters (Sarstedt, Nümbrecht, Germany). In order to purify the His_6_-tagged protein, the Protino^®^ Ni-TED 2000 purification kit was used as recommended by the manufacturer (Macherey and Nagel, Düren, Germany) with modifications. The equilibration of the columns and the washing steps were performed with 50 mM Tris buffer (pH 8.0) containing 250 mM NaCl, followed by three elution steps with 50 mM Tris, 250 mM NaCl, and 250 mM imidazole.

To concentrate the protein, eliminate free P traces, remove imidazole, and reduce NaCl concentration, four subsequent rounds of ultrafiltration using 50 mM Tris buffer (pH 8.0) were conducted by using Vivaspin concentrators, as recommended by the manufacturer (exclusion limit 10 kDa; Sartorius AG, Göttingen, Germany). To remove the thioredoxin tag, fusion proteins were digested by a light chain enterokinase (NEB, Ipswich, MA, USA) (1.5 U/mg fusion protein) at 37 °C for 16 h in buffer (20 mM Tris-HCl, 2 mM CaCl_2_, 50 mM NaCl, pH 8.0). Further purification was performed via hydrophobic interaction chromatography by using a 15PHE 4.6/100PE Tricorn high performance column (bed volume 1.7 mL, flowrate 2 mL/min) and an ÄKTA FPLC system (GE Healthcare, Uppsala, Sweden). Elution was achieved by a linear decreasing NaCl gradient from 1 M to 0 M in 50 mM Tris-HCl (pH 8).

Phosphatase activity was determined at 355 nm by detecting the release of inorganic phosphorous according to the ammonium molybdate method developed by Heinonen and Lahti with modifications [68,69]. The purified enzyme solution (10 μL) was pre-incubated for 3 min at 40 °C in 380 μL of 50 mM sodium acetate buffer (pH 5). Subsequently, 10 μL of 100 mM phytic acid dipotassium salt (Sigma-Aldrich, Munich, Germany) was added, and the mixture incubated for 30 min at 40 °C. To stop the reaction, 1.5 mL of freshly prepared AAM solution (acetone–5 N H_2_SO_4_–10 mM ammonium molybdate) and 100 μL of 1 M citric acid were added. Samples were measured against blanks prepared by adding AAM solution prior to the addition of enzyme. The absorbance (355 nm) was measured using the Ultrosprec^®^ 3300 pro (Amersham plc, Little Chalfont, United Kingdom). All measurements were performed in triplicate. To calculate the enzyme activity, a calibration curve was generated in the range of 5 to 600 nmol phosphate. One activity unit (U) represented the release of 1 nmol phosphate release per min.

The influence of temperature on enzymatic activity was determined via the above-described standard phytase assay. The enzymatic activity was evaluated in a temperature range between 5 and 70 °C by using a temperature-adjusted buffer (50 mM sodium acetate, pH 6). In order to analyze the pH dependence of enzyme activity, the following overlapping buffers were prepared as described by Gomori 1955 [70]: 50 mM glycine-HCl (pH 2.0, 3.0, and 3.6), sodium acetate (pH 3.6, 4.0, 5.0, 5.6, and 6.0), MOPS (pH 6.0, 7.0, and 7.6), Tris-HCl (pH 7.6, 8.0, and 9.0), and glycine-NaOH (pH 9.0).

The substrate specificity was determined using the standard phytase assay under optimal temperature and pH. Ten different substrates (ADP, ATP, NADP, glucose-6-phosphate, glycerophosphate, pyridoxal phosphate, pyrophosphate, naphthyl phosphate, *p*-nitrophenylphosphate, and InsP_6_) were tested at 10 mM. Furthermore, the effect of cations (Al^3+^, Ca^2+^, Cu^2+^, Co^2+^, Fe^2+^, Mg^2+^, Mn^2+^, and Zn^2+^) and the potential inhibitors ethylenediaminetetraacetic acid (EDTA), phenylmethylsulfonyl fluoride (PSMF), wolframate, oxalate, sodium dodecyl sulfate (SDS), and dithiothreitol (DTT), at concentrations of 0.1 and 1 mM, were analyzed.

Kinetic parameters *K_m_* and *V_max_* for Pho16B were calculated from the Michaelis–Menten equation fitted to a non-linear, least-squares regression by using the kinetics module of the program SigmaPlot 12.0 (Systat Software, Inc., San Jose, CA, USA). All measurements were performed under optimal pH and temperature conditions using InsP_6_, *p*-nitrophenylphosphate and naphthyl phosphate as substrates.

### 3.4. Accession Number

The nucleotide sequence of the pLP16 insert sequence has been submitted to the National Center for Biotechnology Information (NCBI) GenBank under accession number KY931681.

## 4. Conclusions

Although PTPs have been relatively well studied, the diversity, role, and characteristics of environmental PTPs remain unknown. All previously described and characterized PTPs were derived from individual microorganisms. Here, for the first time, we characterized an environmentally and metagenome-derived PTP (Pho16B). Pho16B exhibits the characteristic motif of ALPs, which are associated with microbial pathogenesis. At the same time, Pho16B is the first environmentally derived PTP capable of using InsP_6_ as substrate and, thus, the first PTP-phy which does not originate from an isolated microorganism. PTP-phys were described one decade ago, but very little is known about their biological role [21]. The capability of Pho16B to use InsP_6_ as substrate under acidic conditions exposes the relevance of soil as a source of interesting new phytases. Environmental phytases have the potential to be used to solve problems such as eutrophication, associated with the presence of phytic acid in subterranean waters and other bodies of water. New phytases derived from natural environments can be used in agriculture for the rational design of pest inhibitors for crop protection. Similar approaches have been successfully used by analyzing PTPs associated with human diseases, such as diabetes and cancer [71]. Moreover, the high degree of sequence conservation between functional environmental PTPs like Pho16B, and enzymes associated to pathogenicity processes, provides support to the idea of environmental homologous genes as precursors of virulence genes found in clinically relevant bacteria [3]. Therefore, the information obtained from new environmental PTPs provides new valuable insights into the origin of this type of molecule.

## Figures and Tables

**Figure 1 genes-10-00101-f001:**
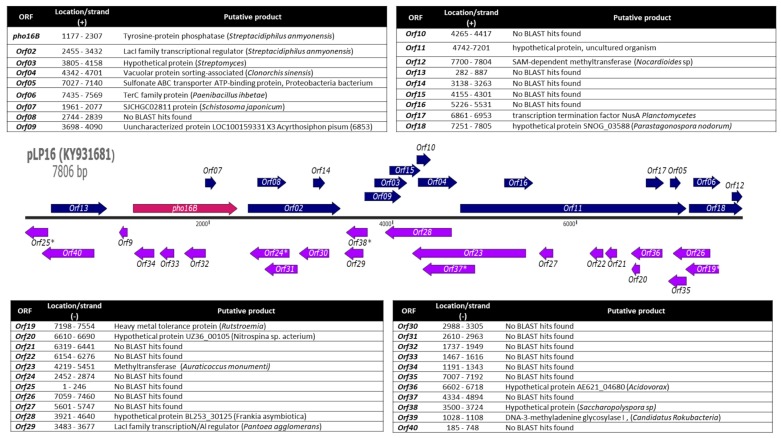
Strand, location, and BLAST results for all identified open reading frames (ORFs) of the pLP16 insert. * Partial ORFs.

**Figure 2 genes-10-00101-f002:**
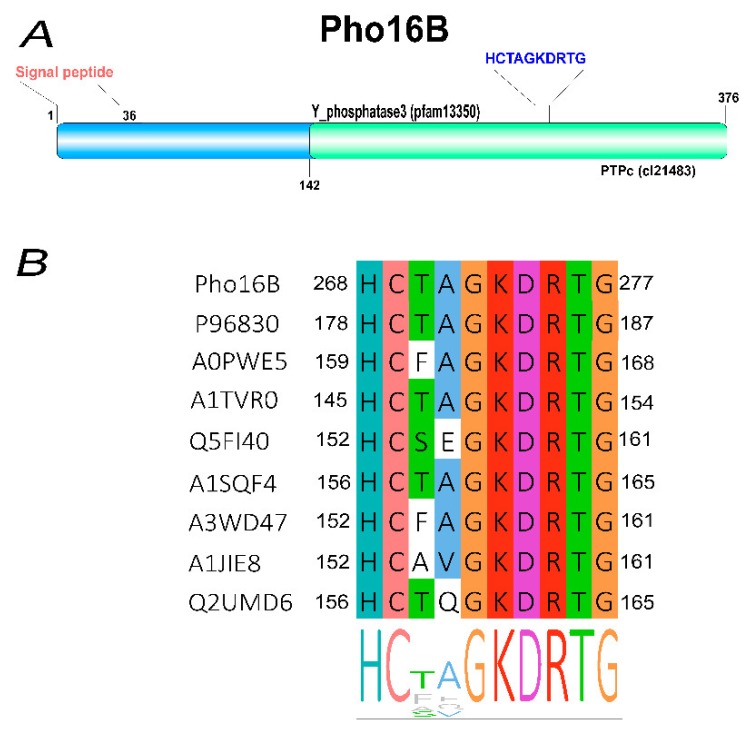
Domain organization and alignment of the subloop in Pho16B (**A**). Architecture of Pho16B showing the positions of the protein tyrosine phosphatase (PTP) domain and the signal peptide. (**B**) Alignment and position of the P-loop motif (HCXXGKDRTG) in Pho16B and other related atypical lipid phosphatase (ALP) proteins. Pho16B (this study), UniProtKB codes: P96830 (*Mycobacterium tuberculosis*), A0PWE5 (*Mycobacterium ulcerans*), A1TVR0 (*Acidovorax citrulli*), Q5FI40 (*Lactobacillus acidophilus*), A1SQF4 *(Nocardioides* sp.), A3WD47 (*Erythrobacter* sp.), A1JIE8 (*Yersinia enterocolitica*), and Q2UMD6 (*Aspergillus oryzae*).

**Figure 3 genes-10-00101-f003:**
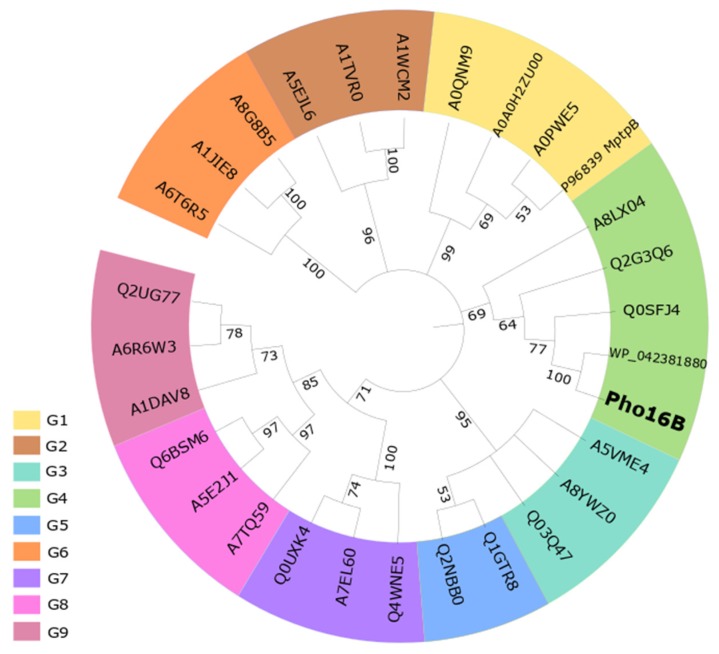
Neighbor-joining phylogenetic tree showing the position of Pho16B. The numbers at the nodes indicate levels of bootstrap support (range from 0 to 100) and were based on 500 replicates. Names correspond to the groups (G) described by Beresford et al. (2010) and their corresponding UniProtKB codes. Pho16B, this study; WP_042381880 (closest related PTP phosphatase from *Streptacidiphilus melanogenes*); P96839 (MptpB *Mycobacterium tuberculosis*); G1: A0QNM9 (*Mycobacterium smegmatis*), A0PWE5 (*Mycobacterium ulcerans*), A0A0H2ZU00 (*Mycobacterium avium*); G2: A1TVR0 (*Acidovorax avenae*), A1WCM2 (*Acidovorax* sp.), A5EJL6 (*Bradyrhizobium* sp.); G3: A5VME4 (*Lactobacillus reuteri*), A8YWZ0 (*Lactobacillus helveticus*), Q03Q47 (*Lactobacillus brevis*); G4: A8LX04 (*Salinispora arenicola*), Q0SFJ4 (*Rhodococcus jostii*), Q2G3Q6 (*Novosphingobium aromaticivorans*); G5: Q1GTR8 (*Sphingopyxis alaskensis*), Q2NBB0 (*Erythrobacter litoralis*); G6: A1JIE8 (*Yersinia enterocolitica*), A6T6R5 (*Klebsiella pneumoniae*), A8G8B5 (*Serratia proteamaculans*); G7: A7EL60 (*Sclerotinia sclerotiorum*), Q0UXK4 (*Phaeosphaeria nodorum*), Q4WNE5 (*Neosartorya fumigata*); G8: A5E2J1 (*Lodderomyces elongisporus*), A7TQ59 (*Vanderwaltozyma polyspora*), Q6BSM6 (*Debaryomyces hansenii*); G9: A6R6W3 (*Ajellomyces capsulatus*), Q2UG77 (*Aspergillus oryzae*), A1DAV8 (*Neosartorya fischeri*).

**Figure 4 genes-10-00101-f004:**
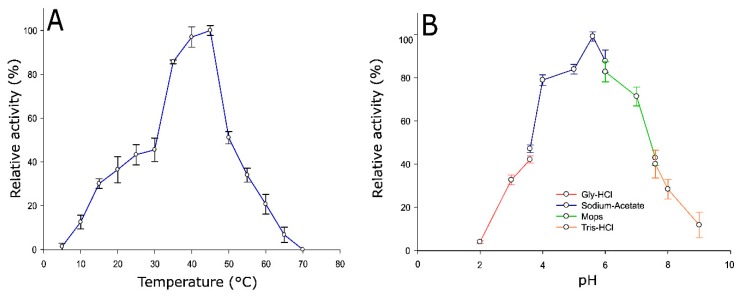
Effect of temperature and pH on Pho16B activity. (**A**) Temperature profile of Pho16B enzymatic activity. (**B**) pH profile of Pho16B enzymatic activity. All measurements were performed in triplicate. Specific activity values are expressed as percentages of the highest relative activity: 8.78 and 4.3 U/mg, for A and B, respectively.

**Figure 5 genes-10-00101-f005:**
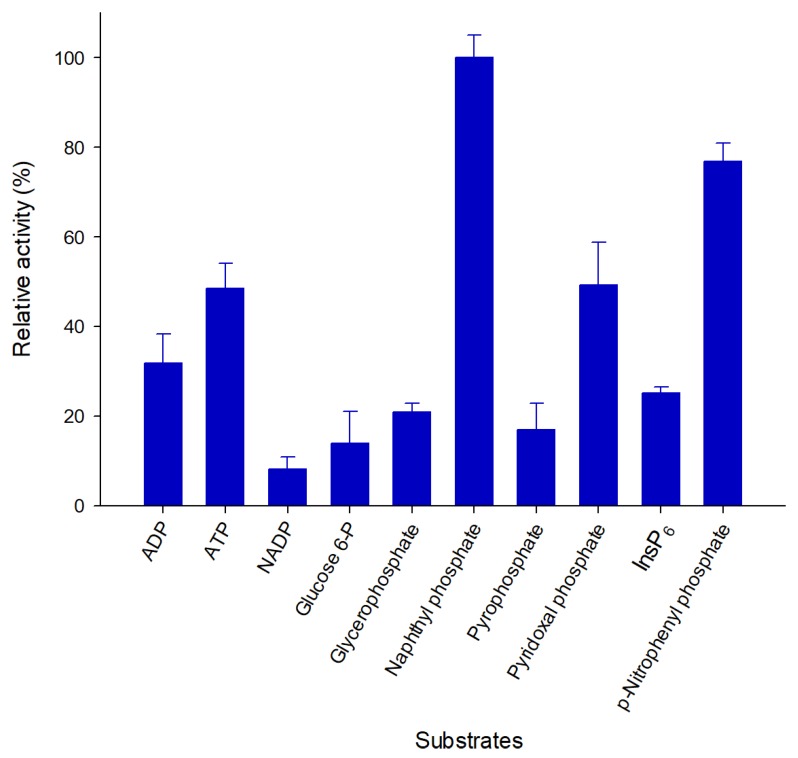
Substrate specificity of Pho16B. Relative activity of Pho16B was measured at 10 mM substrate concentration. All measurements were performed in triplicate and under optimal pH and temperature conditions for enzyme activity. Specific activity values are expressed as percentages of the highest relative activity (13.89 U/mg).

**Figure 6 genes-10-00101-f006:**
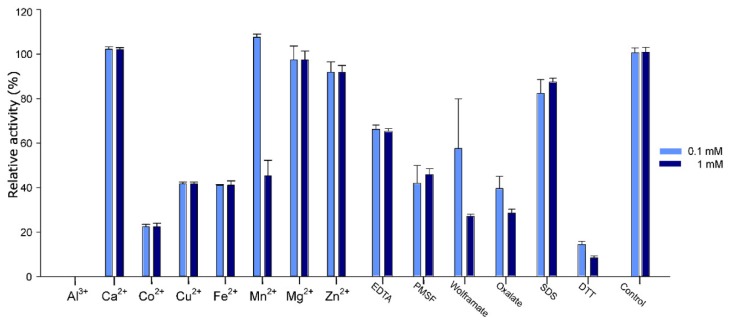
Effect of different concentrations of metal ions and inhibitors on the activity of Pho16B. All measurements were performed in triplicate and under optimal pH and temperature conditions for the enzyme. Specific activity values expressed as percentages of the control reactions (no additions) 8.2 U/mg for both concentrations.

**Table 1 genes-10-00101-t001:** Kinetic values of Pho16B under optimal pH and temperature conditions. All measurements were performed in triplicate.

Substrate	*K_m_* (mM)	*k_cat_* (min^−1^)	*K_cat_*/*K_m_* (s^−1^mM^−1^)
InsP6	1.290 ± 0.38	5.48 ± 0.7	70.43 ± 4.4
Naphthyl phosphate	0.966 ± 0.18	14.64 ± 1.52	238.73 ± 14.52
*p*-Nitrophenylphosphate	1.026 ± 0.14	19.73 ± 2.8	316.89 ± 32.1

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
