# Peer review of "Characteristics of the First Protein Tyrosine Phosphatase with Phytase Activity from a Soil Metagenome"

_genes, 2019, doi:10.3390/genes10020101_

Round 1

Reviewer 1 Report

This manuscript uses the techniques of Functional Metagenomics to obtain a clone or clones that express the new phytase activity. In this, they had some success, with identification of the gene for Pho16B. In taking this approach to getting “new” genes, there is always the chance of isolating some that are not wholly novel and that appears to be the case here, since a similar type is already known that many cultured bacteria.

Right from the start, in the Abstract I read that “Protein tyrosine phosphatases (PTPs) fulfil multiple key regulatory functions”  . And I read elsewhere that “Protein tyrosine phosphatases are a group of enzymes that remove phosphate groups from phosphorylated tyrosine residues on proteins. And yet this is clearly not the sort of enzyme activity that is being sought here. So I was a bit confused. Actually, I did find the Introduction was too long and was rather hard to follow overall and I did not think that the repeated mentions of pathogens seemed to be fully justified.

I thought that they started the Results section with what seemed to be another sort of mini-Introduction, which was not really needed.

They then describe the isolation of the cloned DNA that forms the basis of the rest of the paper. I see that this was obtained from a pre-existing library that they made, but the relative paper (reference 36) is not yet available. So, we do not know what the number of clones in the library was, what was the insert size and what was the vector. These did not appear in the Results or in the Methods section. I also thought that they could have described, very briefly, how they screened the library for the activity in the E. coii colonies.

They then describe the make-up of the particular cloned insert and find many open reading frames.  What seemed very odd was that there were predicted ORFs on both stands for nearly the whole insert. They do not make comments on this, though. I think that they should. I also think that that they could have had an extended Table – probably in the Supplementary Materials – in which they could include the sizes of the predicted proteins and the level of sequence similarity to the nearest homologues in other organisms. In this connection I did not understand why all the ORFs shown were not most closely related to those in the Actinomycete S. melanogenes, since they say that “The taxonomic classification of the complete pLP16 insert revealed complete sequence identity with a genomic region of the Gram-positive soil bacterium Streptacidiphilus melanogenes”. Interestingly, this strain too came from a pine forest.

I thought it was interesting also that the enzyme was predicted to have an N-terminal leader, suggesting that it was extracellular and therefore likely to act on exogenous substrates. However, from what I could see, they never tried to check this directly by getting a culture of S. melanogenes from Korea. I think that they should. Also, I was not sure if their work on the purified protein was on the truncated form or the whole thing, including the signal sequence. Presumably, this extra material at the N-terminus might have an effect on activity. I see (line 346) that they extracted the protein from E. coli cell pellets, so it seems that they did not work on the processed form. One possibility would be to use a vector that works in Corynebacterium or Streptomyces. I noted that when I ran some BLAST comparisons, the closest homologues to the enzyme in was found in a range of Streptomyces species – most of which are, of course, not pathogens. I wonder too why they did not search the increasing numbers of other metagenomic data sets – these might be informing. (In fact I saw that there are many low-level homologues in some such metagenomes, though the matches did not include the N-terminal domains.). That makes me ask another question and that is “Do all the homologues in other known bacteria have N-terminal domains and signal sequences”?

So, this paper is not without interest, and I hope that the authors can submit a shorter crisper MS that deals with the points that are made in the above comments.

Author Response

Right from the start, in the Abstract I read that “Protein tyrosine phosphatases (PTPs) fulfil multiple key regulatory functions”. And I read elsewhere that “Protein tyrosine phosphatases are a group of enzymes that remove phosphate groups from phosphorylated tyrosine residues on proteins”. And yet this is clearly not the sort of enzyme activity that is being sought here. So I was a bit confused. 

Rebuttal: 

PTPs are very diverse (different classes) and act on multiple substrates as they have several functions. The one function of PTPs mentioned by the reviewer (“Protein tyrosine phosphatases are a group of enzymes that remove phosphate groups from phosphorylated tyrosine residues on proteins”)is probably inferred from the Wikipedia entry for PTPs. The Wikipedia entry is incomplete and outdated as only the removal of phosphate groups from phosphorylated tyrosine residues of proteins is mentioned. Some of the most recent reviews regarding PTPs mention other substrates and remark that for many PTPs their substrates remain unknown (for references see list below). This work is focused on the first environmental derived PTP using phytate as substrate (Pho16B). This protein at the same time shares a very specific catalytic signature with PTPs considered as virulence factors. Nevertheless, following the reviewer comment, we try to clarify the matter by adding some examples of the PTP functions and substrates.

References:

Tiganis, T.; Bennett, Anton M. Protein tyrosine phosphatase function: The substrate perspective. Biochem J 2007, 402, 1-15. doi:10.1042/bj20061548

Tonks, N.K. Protein tyrosine phosphatases: From genes, to function, to disease. Nat Rev Mol Cell Biol 2006, 7, 833. doi:10.1038/nrm2039.

Beresford, N.J.; Saville, C.; Bennett, H.J.; Roberts, I.S.; Tabernero, L. A new family of phosphoinositide phosphatases in microorganisms: Identification and biochemical analysis. BMC Genomics 2010, 11, 457-457. doi:10.1186/1471-2164-11-457.

Beresford, N.; Patel, S.; Armstrong, J.; Szöor, B.; Fordham-Skelton, Anthony P.; Tabernero, L. Mptpb, a virulence factor from Mycobacterium tuberculosis, exhibits triple-specificity phosphatase activity. Biochem J 2007, 406, 13-18. doi:10.1042/BJ20070670.

Puhl, A.A.; Greiner, R.; Selinger, L.B. Kinetics, substrate specificity, and stereospecificity of two new protein tyrosine phosphatase-like inositol polyphosphatases from selenomonas lacticifex. Biochem Cell Biol 2008, 86, 322-330. doi:10.1139/O08-095

Actually, I did find the Introduction was too long and was rather hard to follow overall and I did not think that the repeated mentions of pathogens seemed to be fully justified.

Rebuttal:

The introduction was shortened and improved. The mentions related to pathogenicity were reduced as suggested by the reviewer. 

I thought that they started the Results section with what seemed to be another sort of mini-Introduction, which was not really needed.

Rebuttal:

The mentioned paragraph was removed as suggested by the reviewer

I see that this was obtained from a pre-existing library that they made, but the relative paper (reference 36) is not yet available. So, we do not know what the number of clones in the library was, what was the insert size and what was the vector. These did not appear in the Results or in the Methods section. I also thought that they could have described, very briefly, how they screened the library for the activity in the E. coiicolonies.

Rebuttal:

The required information (number of clones of the library, vector, insert size and a brief description of the screening strategy) was added to the methods section. (Lines 387-393).

They then describe the make-up of the particular cloned insert and find many open reading frames. What seemed very odd was that there were predicted ORFs on both stands for nearly the whole insert. They do not make comments on this, though. I think that they should.

Rebuttal:

The corresponding explanation was added in the methods section (lines 397-400).

I also think that that they could have had an extended Table – probably in the Supplementary Materials in which they could include the sizes of the predicted proteins and the level of sequence similarity to the nearest homologues in other organisms.

Rebuttal:

The table with the suggested information was added as new supplementary material (Table S1.)

In this connection I did not understand why allthe ORFs shown were not most closely related to those in the Actinomycete S. melanogenes, since they say that “The taxonomic classification of the complete pLP16 insert revealed complete sequence identity with a genomic region of the Gram-positive soil bacterium Streptacidiphilus melanogenes”. Interestingly, this strain too came from a pine forest.

Rebuttal:

Sorry for this misleading statement, which results in confusion of the reviewer. The statement gave the wrong impression that the cloned DNA fragment was identical to a genomic region of Streptacidiphilus melanogenes. This is not the case. We entirely rephrased the paragraph. The tool used for the taxonomic classification is Kaiju. This is a tool for sensitive taxonomic classification of reads from metagenomic and other data by employing protein level-based taxonomic assignments. By using protein level-based phylogenetic classification, Kaiju performs a more sensitive analysis than methods based on nucleotide comparison (Menzel et al. 2016). 

Based on this protein level-type of taxonomic assignment for example the phytase/phosphatase is related to Streptacidiphilus melanogeneswhereas the overall insert is associated to a relative of this type of organism. However, this does not mean that pLP16 belongs to that particular species. The result has to be interpreted as an indication of the putative taxa associated to the cloned metagenomic material. However, we understand and apologize because in the way the paragraph was written misleads the reader. Therefore, we modified the paragraph and include in the methods section the settings used for Kaiju to obtain this result. Thus, it is not surprising that the individual analysis of the ORFs matches different microorganisms in some cases with relative low sequence identity. 

Reference: Menzel P, Ng KL, Krogh A.2016.Fast and sensitive taxonomic classification for metagenomics with Kaiju. Nat Commun 7:11257.

I thought it was interesting also that the enzyme was predicted to have an N-terminal leader, suggesting that it was extracellular and therefore likely to act on exogenous substrates. However, from what I could see, they never tried to check this directly by getting a culture of S. melanogenes from Korea. I think that they should.

Rebuttal:

As we mentioned above, we cannot say that S. melanogenesis the biological source of pho16B. The taxonomic classification of the whole insert and the individual Blast analysis of several products pointed out in the direction of actinobacterial relative. The product Pho16B reaches only 79% of protein sequence similarity with S. melanogenes. Thus, it is unlikely that the gene is derived from the published S. melanogenes strain/genome. Therefore, the suggested experiment is not indicated as it involves a strain besides other things that is probably not the original host of the metagenomic DNA fragment 

Also, I was not sure if their work on the purified protein was on the truncated form or the whole thing, including the signal sequence. Presumably, this extra material at the N-terminus might have an effect on activity. I see (line 346) that they extracted the protein from E. coli cell pellets, so it seems that they did not work on the processed form. One possibility would be to use a vector that works in Corynebacterium or Streptomyces. I noted that when I ran some BLAST comparisons, the closest homologues to the enzyme in was found in a range of Streptomyces species – most of which are, of course, not pathogens.

Rebuttal:

During the preliminary stages of this research, we used and cloned the truncated form of the gene/enzyme Pho16B (eliminating the putative signal peptide during cloning) for expression experiments using a different expression system. We did not detect differences in the protein activity and substrate spectrum during the preliminary tests in crude extracts. However, the expression system using the truncated form was not suitable for protein purification and characterization as expression of the truncated gene in higher levels in the cytoplasm resulted in unpredictable cell growth and protein production due to impaired cellular growth and extremely low and unpredictable protein expression. We speculate that the presence of Pho16B in the cytoplasm can dephosphorylate key substrates or regulators important for cell growth. 

The aim of this work is focused on reporting the first metagenomic PTP with phytase activity sharing structural characteristics with the ALPs. We did not see the need of changing the expression host, because we were able to probe the phytase activity of Pho16B under the used conditions. Moreover, the closest similar protein of a Streptomyces strain to Pho16B shows only 60% of sequence identity, which might not be enough for correct processing of the protein. 

I wonder too why they did not search the increasing numbers of other metagenomic data sets – these might be informing. (In fact, I saw that there are many low-level homologues in some such metagenomes, though the matches did not include the N-terminal domains.). 

Rebuttal:

Following the comment, we include the results of the search using metagenomic datasets from the NCBI and the EMBL. The results were incorporated into the revised text in lines 129-135. In addition, in the case of the metagenomic EMBL match the best hit possesses also a putative signal peptide.

That makes me ask another question and that is “Do all the homologues in other known bacteria have N-terminal domains and signal sequences”?

Rebuttal:

No, the presence of N-terminal domains and signal peptides varies from organism to organism. There are multiple PTPs with no signal peptide. However, there are also PTPs (phytases and non-phytases) with varying degrees of similarity to Pho16B containing predicted signal peptides e.g. (WP_012939136.1, ABC69367.2, AAQ13669.1, Q2G3Q6, WP_006305978., CAE79111.1, WP_079138152.1. In fact, as we mentioned above the sequence MGYP000356208135 (best match of Pho16B against metagenomic data) harbors a signal peptide. 

Reviewer 2 Report

The manuscript by Villamizar and colleagues details the identification and characterization of a new protein tyrosine phosphatase (Pho16B) derived from a soil metagenomic library. The authors provide a thorough description of the sequence analysis, structural predictions, and activity assays under conditions of varying temperature, pH, additives, and with ten different substrates. This is an important study describing the first environmentally-derived PTP, providing critical experimental data that will help inform future study of PTPs across environmental and human-associated systems. The work is well done and the manuscript written in a concise and thoughtful manner. I have only a few minor comments below.

L. 34: “OMICS” does not need to be capitalized.

L. 91: Replace “.” with “,”.

L. 102: Replace “able” with “capable”.

Fig. 3: Suggest labeling on the figure the phosphatase groups G1-G9, along with removing the gray shading on the phylogenetic tree and increasing the font size for the bootstrap support values.

L. 407: Should read: “…PTP-phy, which does not originate from an isolated microorganism.”

L. 414: Should read: “… associated with human diseases…”

Author Response

Reviewer 2

L. 34: “OMICS” does not need to be capitalized.

Rebuttal:

Changed as recommended by the reviewer

L. 91: Replace “.” with “,”.

Rebuttal:

Changed as recommended by the reviewer

L. 102: Replace “able” with “capable”.

Rebuttal:

Changed as recommended by the reviewer

Fig. 3: Suggest labeling on the figure the phosphatase groups G1-G9, along with removing the gray shading on the phylogenetic tree and increasing the font size for the bootstrap support values.

Rebuttal:

Figure adapted according to the suggestion of the reviewer. 

L. 407: Should read: “…PTP-phy, which does not originate from an isolated microorganism.”

Rebuttal:

Changed as recommended by the reviewer

L. 414: Should read: “… associated with human diseases…”

Rebuttal:

Changed as recommended by the reviewer

Round 2

Reviewer 1 Report

The authors have answered nearly all my original comments in a satisfactory way and, in the other cases, have given reasonable explanations why they cannot or need not accommodate these issues.